# Multimodal Lego: Model Merging and Fine-Tuning Across Topologies and Modalities

**Konstantin Hemker**
Department of Computer Science & Technology
University of Cambridge
Cambridge, United Kingdom
konstantin.hemker@cl.cam.ac.uk

**Nikola Simidjievski**
Department of Oncology
University of Cambridge
Cambridge, United Kingdom
ns779@cam.ac.uk

**Mateja Jamnik**
Department of Computer Science & Technology
University of Cambridge
Cambridge, United Kingdom
mateja.jamnik@cl.cam.ac.uk

## Abstract

Learning holistic computational representations in physical, chemical or biological systems requires the ability to process information from different distributions and modalities within the same model. While there are many available multimodal fusion and alignment approaches, most of them require end-to-end training, scale quadratically with the number of modalities, cannot handle cases of high modality imbalance in the training set, or are highly topology-specific, making them too restrictive for many biomedical learning tasks. This paper presents *Multimodal Lego* (MM-Lego), a general-purpose fusion framework to turn any set of encoders into a competitive multimodal model with no or minimal fine-tuning. We achieve this by introducing a wrapper for any unimodal encoders that enforces shape consistency between modality representations and harmonises these representations by learning features in the frequency domain to enable model merging with little signal interference. We show that MM-Lego 1) can be used as a *model merging* method which achieves competitive performance with end-to-end fusion models *without any fine-tuning*, 2) can operate on any unimodal encoder, and 3) is a *model fusion* method that, with minimal fine-tuning, achieves state-of-the-art results on six benchmarked multimodal biomedical tasks.

## 1 Introduction

The utility and demand for multimodal machine learning approaches has sharply risen due to their potential to derive holistic representations in various systems, including physics [1], chemistry [2], neuroscience [3], or biology [4]. Multimodal models in the vision & language domains leverage the same data distributions, which are represented across different modalities [5, 6, 7], such as vision-text pairs of the same concepts. However, in many biomedical domains, modalities represent data at different scales (e.g., cellular, genomic, transcriptomic, etc.), cardinalities that are not paired (e.g., many single-cell reads for a single tissue slide per patient), and follow separate distributions. While large foundation models have excelled in tasks confined to individual modalities [8, 9, 10], training these models across modalities is an expensive end-to-end process, that requires paired modalities. One recently emergent solution to these challenges is presented through *model merging* [11] (also referred to as *knowledge fusion* [12]), an approach commonly used in the context of multi-task settings and language

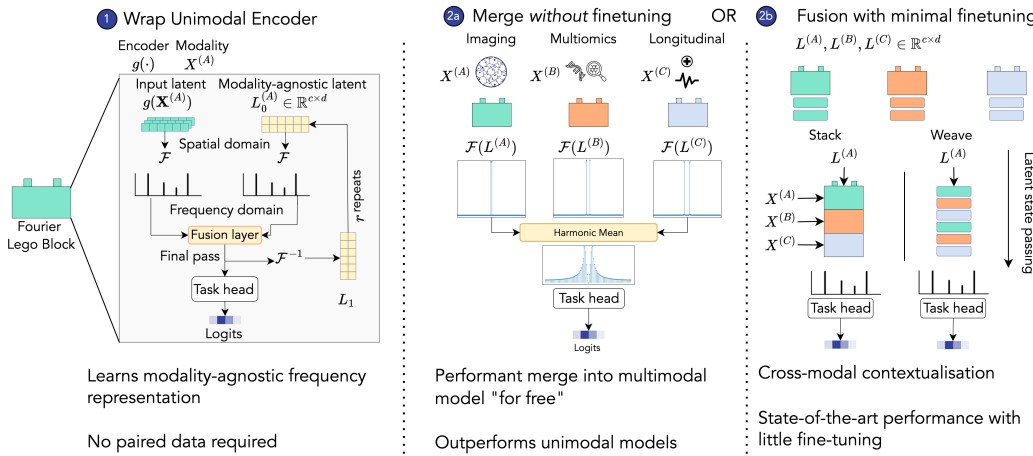

Figure 1: The Multimodal Lego workflow to turn a set of encoders into a performant multimodal model. *LegoBlock* (1) makes unimodal encoders compatible with model merging techniques by learning a latent representation in the frequency-domain to prevent signal interference effects upon aggregation. Any set of *LegoBlocks* can be merged into a multimodal model without any fine-tuning (*LegoMerge* (2a)) or with minimal fine-tuning to achieve state-of-the-art performance (*LegoFuse* (2b)).

modelling, which capitalises on combining well-performing unimodal models trained in isolation. Model merging methods attempt to combine two architecturally identical models trained on different distributions through interpolation, arithmetic manipulation and aggregation of their weights [13, 14, 15], or stacking their layers [16], often without additional training/fine-tuning. While model merging has been extended to some multimodal vision and language tasks [17], its crucial challenges in a multimodal setting are that: a) the merged components are still trained in isolation, and b) we cannot assume topological equivalence between two models for separate modalities due to their separate input shapes.

In this paper, we present Multimodal Lego (*MM-Lego*) – a flexible framework for combining various unimodal models into a multimodal model with no or minimal fine-tuning (Figure 1). We introduce two approaches within our framework – *LegoFuse* and *LegoMerge*, enabling performant multimodal models given a set of unimodal encoders with either little (*LegoFuse*) or no (*LegoMerge*) fine-tuning. We show that MM-Lego satisfies multiple desirable properties in a range of real-world multimodal applications combining imaging, tabular, and time series modalities. We demonstrate the utility of MM-Lego on seven medical datasets across three separate downstream tasks, showing that it is: 1) performant without end-to-end training, 2) topology agnostic, 3) is scalable, and 4) handles modality imbalance and non-overlapping sets.

**Fusion methods.** Given multiple data inputs or latent representations, fusion methods construct single learning representations that can be used for a downstream task, often whilst reducing dimensionality. Many fusion methods [3, 18] first learn a set of modality-specific encoders $\mathcal{G} = \{g_m : m \to \mathbf{h}^{(m)}\}$ assuming a single task label $\mathbf{y}$. This results in a set of latent representations $\mathcal{H} = \{g_m(m), m \in \mathcal{M}\}$, which are combined with a fusion operator to obtain the final fused representation $\mathbf{z} = \psi(\mathcal{H})$ and its final prediction $\hat{\mathbf{y}} = f(\mathbf{z})$. Following this problem setup, fusion methods can be differentiated by: 1) the choice of the *fusion operator* $\psi(\cdot)$; 2) the *fusion stage* in the pipeline of when $\psi(\cdot)$ is applied; and 3) the *fusion order* in which the fusion operations are applied (i.e., sequential vs. parallel). The *fusion operator* can be either static (e.g., concatenation [19], Kronecker product [20]) or learnable (e.g., low-rank tensor fusion [21], cross-attention mechanisms [22, 23, 24], mixture of experts [25, 26]). The *fusion stage* is typically characterised as early, intermediate or late fusion. Early fusion methods apply a static fusion operator $\psi(\cdot)$ to the raw data while only applying this after passing each modality through $\mathcal{G}$. Intermediate fusion methods often don't apply a static aggregation but rather learn a fusion function (i.e., a small sub-network or neural layer) in the latent space as part of its end-to-end training [27].

**Model merging.** The core idea behind model merging, typically deployed in multi-task settings, is that earlier layers in a network may learn similar features that may be used across tasks [28]. Using linear interpolation [11] or arithmetic manipulation [15] of the task-specific weights, model merging approaches have shown that they can effectively generalise to new tasks without any fine-tuning.

Formally, given multiple tasks $\mathbf{y}^{(\mathcal{T})}$ for the same modality $A$, they first learn the set of task-specific functions $\mathcal{F} = \{f_t(\omega_t) : A \to \hat{\mathbf{y}}^{(t)} \mid t \in \mathcal{T}\}$ where $\omega$ denotes the corresponding model parameters. Assuming the same architecture for each model in $\mathcal{F}$, parameters from a pre-trained base model $\omega_{base}$ can be used to derive task vectors as $\mathcal{V} = \{\tau_t \leftarrow \omega_t - \omega_{base} \mid t \in \mathcal{T}\}$ [15] . Given these task vectors, a multi-task model can be constructed by updating the weights of the base model $\omega' = \omega_{base} + \lambda \sum_t^{\mathcal{T}} \tau_t$. This idea has been extended by the TIES [13] and DARE [29] that merge models through sparsifying and resolving sign conflicts in the task vectors. Another popular approach is spherical linear interpolation (SLERP), a method used to smoothly interpolate between two vectors while respecting the geometric properties of the vector space [11]. More specifically, given model parameters $\omega_{T_1}$ and $\omega_{T_2} \in \mathbb{R}^d$, derived from two models with identical architectures, the merged multi-task model parameters can be calculated as $\omega' = \omega_{T_1} \frac{sin(\theta \cdot (1-\mu))}{sin(\theta)} + \omega_{T_2} \frac{sin(\theta \cdot \mu)}{sin(\theta)}$ where $\theta$ is the radian between the two vectors $\omega_{T_1}$ and $\omega_{T_2}$ [30]. The underlying assumption of the above model merging approaches is that the models should have an equivalent network topology, ensuring that the dimensions $\mathbb{R}^d$ match up between tasks. However, while this is an acceptable constraint for multi-task learning, it is infeasible for multimodal models where modality shapes and the corresponding network topologies vary greatly.

## 2 Multimodal Lego

**Preliminaries.** Let $\mathbf{X}^{(\mathcal{M})} = \bigcup_{m \in \mathcal{M}} x_m$ be a multimodal dataset where $\mathcal{M} = \{A, B, \dots\}$ represents the set of modalities $m$ such as images ($A$), tabular data ($B$), time series ($C$), etc. Let $\mathbf{X}^{(A)}_{i,j,k}$ correspond to the element in the dataset for modality $A$ at sample $i$, column $j$, and channel $k$, assuming $A \in \mathbb{R}^{I \times J \times K}$ where $1 \le i \le N$, $1 \le j \le J$, $1 \le k \le K$. Each sample in $\mathbf{X}$ has a set of discriminative task labels $\mathbf{y}^{(\mathcal{T})} = \bigcup_{t \in \mathcal{T}} \mathbf{y}^{(t)}$, where $\mathcal{T} = \{T_1, T_2 \dots, T_c\}$ is the set of possible tasks such that $\mathbf{y}^{(T_1)} = \{y_1^{T_1}, y_2^{T_1}, ..., y_N^{T_1}\}$ are the scalar target values for task $T_1$ for $N$ samples.

**Architecture.** Rather than learning a single fusion operator $\psi(\mathcal{H})$ that applies to all latent representations at once, we learn a set of latent update functions for each modality, in the form of

$$\mathcal{B} = \{\psi_m : (g_m(X^{(m)}), L_s^{(m)}) \to L_{s+1}^{(m)} \mid s \in S, m \in \mathcal{M}\}, \tag{1}$$

where $L_t^{(m)} \in \mathcal{L}$ is our target latent representation for each modality that we will later use in the merge and fusion, and $S \in \mathbb{Z}^+$ is the number of update steps.

Using the iterative update architecture with latent state passing in Equation 1 has a number of advantages. First, iterative attention architectures have been shown to be highly generalisable across modalities [31], and effective at dealing with missing individual modalities [32, 33]. Second, since all modalities are encoded into a self-defined latent representation, we can impose a dimensionality constraint such that each latent has the same dimensions (e.g., $L^{(A)}, L^{(B)}, L^{(C)} \in \mathbf{R}^{c \times d}$ for latent channels and dimensions $c$ and $d$). Third, we can do latent state passing between elements in $\mathcal{B}$, which allows us to "stack" the update functions on top of each other (hence the name) to sequentially encode each modality's signal into the same latent representation.

**LegoBlocks.** Each element in $\mathcal{B}$ represents a *LegoBlock*, which learns the latent update function $\psi_m$ for any given encoder $g_m$. Acknowledging that different data modalities and structures require different inductive biases to effectively encode each modality's information ($g_m$), *LegoBlock* acts as a wrapper to accurately encode its latent $h_m$ into $L^{(m)}$. The benefit of training each modality update function separately instead of end-to-end is that we can train on entirely separate samples for the same tasks. For example, in many medical domains, we may have single-cell data for one subset of patients and bulk sequencing data for a different subset, while having the same task labels for the entire set. To address this, we use a latent representations $L$ that effectively encode signal across modalities, and are robust or invariant to transformations (shifts, rotations, etc.), noise and signal interference.

This motivated us to design MM-Lego for learning latent representations in the frequency domain, taking advantage of a number of desirable properties for multimodal merging and fusion. In particular, frequency-domain representations are: 1) *signal-preserving* as frequency features are less prone to signal interference upon aggregation (see Appendix F); 2) *distance-preserving*, as the Euclidean distance between two signals remains unchanged after the Fourier Transform (following from

Parseval's Theorem [34]), making them suitable for distance-based loss functions; 3) *invertible* as the spatial/temporal domain can be reconstructed, allowing for the iterative updates outlined in Equation 1; and 4) *efficient*, as the Fast Fourier Transform (FFT) has a time complexity of $O(n \log(n))$, making it scalable to very large datasets [35].

Starting with the latent representation in the spatial domain, we first apply a discrete FFT $\mathcal{F}(\cdot)$ [36] along each dimension of the 2D Tensor to yield a frequency domain representation. $L_t^{\mathcal{F}}(u,v) = \sum_{i=0}^{c-1} \sum_{j=0}^{d-1} L_t(i,j)e^{-2\pi i(\frac{ux}{c} + \frac{vy}{d})}$, where $i, j$ denote the spatial-domain indices, and $u, v$ denote the frequency-domain indices. This results in a complex frequency-domain representation from which we separate the real (symmetrical) and imaginary (asymmetrical) components of the FFT $((L_t^{\mathcal{F}})^r$ and $(L_t^{\mathcal{F}})^i)$ [37]. We update the real component using a standard cross-attention layer [38], where we aim to learn the weight matrices $W_m^q$ for the update query $(L_t^{\mathcal{F}})^r$, and $W_m^k$, $W_m^v$ for the keys and values $(h^{(A)})$ resulting in the latent update:

$$(L_{t+1}^{\mathcal{F}})^r = \text{softmax}\left(\frac{(L_t^{\mathcal{F}})^r W_m^q \cdot (h^{(A)} W_m^k)^\top}{\sqrt{d_k}}\right) \cdot (h^{(A)} W_m^v). \tag{2}$$

In contrast to other Fourier-based architectures [35], which only use the real component of the transform, we keep track of the imaginary component $(L_t^{\mathcal{F}})^i$ as well. This allows us to reconstruct the complex representation, and subsequently apply the inverse transform. We found this to be critical for our iterative architecture, as otherwise the signal gets distorted and we lose phase information (encoded in the imaginary component) at each update pass. Once we reconstruct the complex representation, we apply the inverse transform to recover the spatial representation in preparation of the next pass $L_{t+1} = \mathcal{F}^{-1}((L_{t+1}^{\mathcal{F}})^r + i(L_t^{\mathcal{F}})^i)$. Finally, the last task-specific heads of each block are a fully-connected layer after applying layer normalisation. We omit the inverse transform after the last update such that each head is trained in the frequency domain. This ensures that we can apply aggregations with low signal interference on $\mathcal{L}$ during *LegoMerge*.

**LegoMerge.** *LegoMerge* constructs a performant multimodal model without any additional training. With the architectural assumptions imposed on each modality encoder in $\mathcal{G}$ through *LegoBlocks* $\mathcal{B}$, we can apply model merging techniques in a multimodal setting. With $\mathcal{L} \subseteq \mathbb{R}^{c \times d}$ and each element in $\mathcal{L}$ being in the frequency domain, we can use aggregation functions $\psi(\cdot)$, which are less prone to cancelling out signal. For example, let $L^{(A)}$ and $L^{(B)}$ be the final frequency domain latent representations for modalities $A$ and $B$, then we can calculate a merged multimodal representation as:

$$\psi(L^{(A)}, L^{(B)}) = \left(\frac{2|L^{(A)}| \cdot |L^{(B)}|}{|L^{(B)}| + |L^{(A)}|}\right) \cdot e^{i \cdot \frac{\angle L^{(A)} + \angle L^{(B)}}{2}}, \tag{3}$$

where the real component is the harmonic mean of the magnitudes ($|\cdot|$), and the imaginary component is the arithmetic mean of the phases ($\angle$) of $L^{(A)}$ and $L^{(B)}$. We take the harmonic mean since it is less prone to outliers [37], that is, the merged representation is less likely to be strongly skewed towards either modality by very large frequency components. With the cross-modal combined representation $L^{(\mathcal{M})}$, we need to combine the task heads of each block, where we apply spherical linear interpolation (SLERP) [30] for the set of task heads $\mathcal{Y}$ from each element in $\mathcal{B}$.

**LegoFuse.** *LegoFuse* overcomes the limitations of LegoMerge of training each element in $\mathcal{B}$ in isolation, thus allowing for modalities to mutually contextualise each other. As such it requires a minimal amount of fine-tuning. To avoid fine-tuning a potentially noised signal emerging from the merged latent $L^{(\mathcal{M})}$, *LegoFuse* operates at the layer level (by sequentially passing through all layers in $\mathcal{B}$), rather than directly fine-tuning the merged model (at the parameter-level). Specifically, the shape consistency introduced by $\mathcal{L} \subseteq \mathbb{R}^{c \times d}$ allows the stacked model to pass the Fourier-transformed latent states either between blocks (stacking) or different layers between blocks (weaving), as illustrated in Figure 1. We then fine-tune the stacked/weaved model for a few epochs with all (paired) modalities, such that the state updates are conditioned on all modalities' updates. This, in turn, becomes the query for the cross-attention layer. Note that, both the stacked and weaved variants of *LegoFuse* allow for fine-tuning all model parameters, including the ones of the initial modality-specific encoders.

## 3 Results & Discussion

**Experiments.** We evaluate MM-Lego (*LegoMerge* and *LegoFuse*) and its components (*LegoBlock*) on seven multimodal medical datasets covering three separate modalities (images, tabular, time series)

Table 1: Comparison of desirable requirements of multimodal systems in medical domains. ✓: meets requirement, (✓): some approaches meet requirement, ✗: fails requirement.

| Criteria/Method | Late | Intermediate | Early | Multi-task merge | LegoMerge | LegoFuse |
|---|---|---|---|---|---|---|
| Performant without end-to-end training | ✗ | ✗ | ✗ | ✓ | ✓ | ✓ |
| Learns cross-modal interactions | ✗ | ✓ | (✓) | ✗ | ✗ | ✓ |
| Architecture agnostic | ✓ | (✓) | ✓ | ✗ | ✓ | ✓ |
| Handles strong modality imbalance | ✗ | (✓) | ✗ | ✓ | ✓ | ✓ |
| Add modalities without re-training | ✗ | ✗ | ✗ | ✗ | ✓ | (✓) |

Table 2: Mean and std. dev. of task performance, showing the concordance Index (survival) and AUC (classification) on 5 random sub-sampling folds with the **best** and second-best models highlighted.

| | BLCA | BRCA | KIRP | UCEC | ICD9 | MORT | ISIC |
|---|---|---|---|---|---|---|---|
| *Samples* | n=436 | N=1021 | n=284 | n=538 | n=32616 | n=32616 | n=2875 |
| *Modalities* | tab, img | tab, img | tab, img | tab, img | tab, ts | tab, ts | tab, img |
| *Metric* | c-Index | c-Index | c-Index | c-Index | AUC | Macro AUC | AUC |
| **UniModal (Tabular)** | | | | | | | |
| SNN [42] | $0.689_{\pm0.012}$ | $0.544_{\pm0.020}$ | $0.798_{\pm0.035}$ | $0.589_{\pm0.057}$ | $0.731_{\pm0.023}$ | $0.634_{\pm0.020}$ | $0.507_{\pm0.005}$ |
| MultiModN [32] | $0.500_{\pm0.000}$ | $0.500_{\pm0.000}$ | $0.525_{\pm0.140}$ | $0.500_{\pm0.000}$ | $0.500_{\pm0.000}$ | $0.500_{\pm0.000}$ | $0.500_{\pm0.000}$ |
| Perceiver [19] | $0.686_{\pm0.009}$ | $0.557_{\pm0.016}$ | $0.836_{\pm0.053}$ | $0.615_{\pm0.035}$ | $0.629_{\pm0.023}$ | $0.658_{\pm0.000}$ | $0.840_{\pm0.084}$ |
| **UniModal (Image/T.Series)** | | | | | | | |
| ABMIL [43] | $0.591_{\pm0.057}$ | $0.610_{\pm0.093}$ | $0.741_{\pm0.080}$ | $0.558_{\pm0.040}$ | $0.614_{\pm0.025}$ | $0.691_{\pm0.014}$ | $0.500_{\pm0.000}$ |
| MultiModN [43] | $0.520_{\pm0.022}$ | $0.527_{\pm0.150}$ | $0.570_{\pm0.156}$ | $0.564_{\pm0.097}$ | $0.500_{\pm0.000}$ | $0.544_{\pm0.033}$ | $0.500_{\pm0.000}$ |
| Perceiver [19] | $0.532_{\pm0.027}$ | $0.604_{\pm0.064}$ | $0.716_{\pm0.063}$ | $0.534_{\pm0.106}$ | $0.700_{\pm0.013}$ | $0.715_{\pm0.016}$ | $0.719_{\pm0.050}$ |
| **MultiModal** | | | | | | | |
| SNN + ABMIL (CC, Late) | $0.561_{\pm0.000}$ | $0.541_{\pm0.104}$ | $0.841_{\pm0.128}$ | $0.601_{\pm0.018}$ | $0.628_{\pm0.020}$ | $0.617_{\pm0.015}$ | $0.661_{\pm0.196}$ |
| SNN + ABMIL (BL, Late) | $0.622_{\pm0.054}$ | $0.557_{\pm0.089}$ | $0.811_{\pm0.108}$ | $0.666_{\pm0.031}$ | $0.500_{\pm0.000}$ | $0.500_{\pm0.001}$ | $0.501_{\pm0.002}$ |
| Perceiver (CC, Early) | $0.547_{\pm0.060}$ | $0.561_{\pm0.105}$ | $0.692_{\pm0.000}$ | $0.548_{\pm0.000}$ | $0.733_{\pm0.028}$ | $0.723_{\pm0.015}$ | $0.721_{\pm0.198}$ |
| MultiModN (Inter.) | $0.524_{\pm0.018}$ | $0.500_{\pm0.000}$ | $0.602_{\pm0.076}$ | $0.512_{\pm0.008}$ | $0.500_{\pm0.000}$ | $0.500_{\pm0.000}$ | $0.500_{\pm0.000}$ |
| MCAT (Inter.) [44] | $0.702_{\pm0.032}$ | $0.564_{\pm0.000}$ | $0.823_{\pm0.076}$ | $0.633_{\pm0.068}$ | $0.500_{\pm0.000}$ | $0.500_{\pm0.000}$ | $0.627_{\pm0.059}$ |
| HEALNet (Inter.) [44] | $0.714_{\pm0.025}$ | $0.618_{\pm0.063}$ | $0.842_{\pm0.063}$ | $0.594_{\pm0.023}$ | $0.767_{\pm0.022}$ | $0.748_{\pm0.009}$ | $0.639_{\pm0.09}$ |
| **LegoMerge** | $0.701_{\pm0.021}$ | $0.601_{\pm0.025}$ | $0.825_{\pm0.114}$ | $0.625_{\pm0.080}$ | $0.684_{\pm0.015}$ | $0.751_{\pm0.027}$ | $0.721_{\pm0.143}$ |
| **LegoFuse, w/ 2 epochs** | $0.734_{\pm0.032}$ | $0.626_{\pm0.046}$ | $0.863_{\pm0.112}$ | $0.634_{\pm0.010}$ | $0.771_{\pm0.020}$ | $0.759_{\pm0.041}$ | $0.701_{\pm0.023}$ |

from three separate sources: histopathology (The Cancer Genome Atlas (TCGA)) [39], intensive care data (Medical Information Mart for Intensive Care (MIMIC)) [40], and skin imaging (International Skin Imaging Collaboration (ISIC)) [41]. The seven tasks shown in our results correspond to survival analysis tasks on four TCGA sites (BLCA, BRCA, KIRP, UCEC), classification tasks on two variants of MIMIC (disease classification (ICD9) and patient mortality (MORT)), and predicting melanoma for the ISIC patients. Further details on datasets and task setup can be found in Appendices C and D.

**Discussion.** With the increasing volume, complexity and diversity of collected biomedical data, (re)training multimodal models from scratch becomes more expensive, unsustainable, and even infeasible. Going beyond computational constraints, further desired properties that guided the design of *MM-Lego* are outlined in Table 1. Our results in Table 2 provide strong evidence that *MM-Lego* meets these requirements, efficiently achieving competitive performance.

> *LegoMerge* matches end-to-end trained multimodal baselines in most tasks without any additional training, while *LegoFuse* outperforms strong baselines with as little as 2 epochs of fine-tuning. Notably, *LegoMerge* does not require a single paired modality training sample whilst still being useful for multimodal inference, outperforming ensemble models (Appendix B). Our results also show that *MM-Lego* addresses a key limitation in model merging literature which assumes topology equivalence. While this is a feasible assumption for model merging in multi-task learning, different data shapes across modalities limit the application of these methods in multimodal settings. Therefore, the design of *LegoBlock* is sufficiently permissive to use any unimodal encoder as part of this framework, whilst enforcing the necessary architectural assumptions required for model merging. Our findings (in Figure 2, Appendix 3) support this, showing that any unimodal encoder (such as SNNs and AMIL) can be wrapped in a *LegoBlock* without any practical loss in performance.

To the best of our knowledge, MM-Lego is the first general-purpose model merging framework for multimodal data outside of the vision & language domains.

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

# A  Notation

**Objects.**

- $X^{(A)}$: matrix corresponding to modality A
- $\mathbf{x}^{(A)}$: a vector in $X^{(A)}$ (e.g., a sample of modality A)
- $\mathbf{X}_{i,j,k}^{(A)}$: elements of matrix $X^{(A)}$ at row $i$, column $j$, channel $k$, assuming $X^{(A)} \in \mathbb{R}^{I \times J \times K}$ where $1 \leq i \leq I, 1 \leq j \leq J, 1 \leq k \leq K$
- $\mathbf{X}^{(\mathcal{M})} = \bigcup_{m \in \mathcal{M}} X^{(m)}$: multimodal dataset
- $\mathbf{y} \in \mathcal{Y} = \bigcup_{t \in \mathcal{T}} \mathbf{y}^{(t)}$: set of task labels for all available tasks $\mathcal{T}$
- $\mathbf{y}^{(T_1)}$: task labels for task $T_1$

**Sets.**

- $\mathcal{M}$: set of modalities
- $\mathcal{T}$: set of tasks
- $\mathcal{Y}$: set of task-specific heads
- $\mathcal{G} = \{g_m : m \to \mathbf{h}^{(m)} \mid m \in \mathcal{M}\}$: set of modality-specific encoders
- $\mathcal{H}_\mathbf{y} = \{g_m(m, \mathbf{y}) \mid m \in \mathcal{M}\}$: set of task- and modality-specific embeddings
- $\mathcal{B} = \{\psi_m : (g_m(X^{(m)}), L_s^{(m)}) \to L_{s+1}^{(m)} \mid s \in S, m \in \mathcal{M}\}$: set of *LegoBlocks*

**Functions and Operators.**

- $g_m(\cdot)$: modality-specific encoder
- $\psi(\cdot)$: fusion operator (monolithic)
- $\psi_m(\cdot)$: modality-specific latent update
- $\mathcal{F}$: Fourier transform
- $\mathcal{F}^{-1}$: Inverse Fourier transform

# B    Full Results

| | BLCA | BRCA | KIRP | UCEC | ICD9 | MORT | ISIC |
|---|---|---|---|---|---|---|---|
| *Samples* | n=436 | N=1021 | n=284 | n=538 | n=32616 | n=32616 | n=2875 |
| *Modalities* | tab, img | tab, img | tab, img | tab, img | tab, ts | tab, ts | tab, img |
| *Metric* | c-Index | c-Index | c-Index | c-Index | AUC | Macro AUC | AUC |
| **Tabular** | | | | | | | |
| SNN | 0.689±0.012 | 0.544±0.020 | 0.798±0.035 | 0.589±0.057 | 0.731±0.023 | 0.634±0.020 | 0.507±0.005 |
| MultiModN | 0.500±0.000 | 0.500±0.000 | 0.525±0.140 | 0.500±0.000 | 0.500±0.000 | 0.500±0.000 | 0.500±0.000 |
| Perceiver | 0.686±0.009 | 0.557±0.016 | 0.836±0.053 | 0.615±0.035 | 0.629±0.023 | 0.658±0.000 | **0.840±0.084** |
| LegoBlock | 0.681±0.015 | 0.591±0.021 | 0.840±0.135 | 0.615±0.031 | 0.645±0.017 | 0.619±0.028 | 0.668±0.141 |
| **Image/Time Series** | | | | | | | |
| ABMIL | 0.591±0.057 | 0.610±0.093 | 0.741±0.080 | 0.558±0.040 | 0.614±0.025 | 0.691±0.014 | 0.500±0.000 |
| MultiModN | 0.520±0.022 | 0.527±0.150 | 0.570±0.156 | 0.564±0.097 | 0.500±0.000 | 0.544±0.033 | 0.500±0.000 |
| Perceiver | 0.532±0.027 | 0.604±0.064 | 0.716±0.063 | 0.534±0.106 | 0.700±0.013 | 0.715±0.016 | 0.719±0.050 |
| LegoBlock | 0.568±0.029 | 0.533±0.000 | 0.630±0.182 | 0.565±0.069 | 0.643±0.013 | 0.711±0.008 | 0.706±0.147 |
| **MultiModal** | | | | | | | |
| **LegoMerge (Ours)** | 0.701±0.021 | 0.601±0.025 | 0.825±0.114 | 0.625±0.080 | 0.684±0.015 | **0.751±0.027** | **0.721±0.143** |
| **Merge Uplift vs. best block** | 2.9% | 1.7% | -1.8% | 1.6% | 5.7% | 5.3% | 2.1% |
| SNN + ABMIL (CC, Late) | 0.561±0.000 | 0.541±0.104 | 0.841±0.128 | 0.601±0.018 | 0.628±0.020 | 0.617±0.015 | 0.661±0.196 |
| SNN + ABMIL (LR, Late) | 0.622±0.054 | 0.557±0.089 | 0.811±0.108 | **0.666±0.031** | 0.500±0.000 | 0.500±0.001 | 0.501±0.002 |
| Perceiver (CC, Early) | 0.547±0.060 | 0.561±0.105 | 0.692±0.000 | 0.548±0.000 | 0.733±0.028 | 0.723±0.015 | **0.721±0.198** |
| MultiModN (Inter) | 0.524±0.018 | 0.500±0.000 | 0.602±0.076 | 0.512±0.008 | 0.500±0.000 | 0.500±0.000 | 0.500±0.000 |
| MCAT (Inter) | 0.702±0.032 | 0.564±0.000 | 0.823±0.076 | 0.633±0.068 | 0.500±0.000 | 0.500±0.000 | 0.627±0.059 |
| HEALNet (Inter) | **0.714±0.025** | **0.618±0.063** | **0.842±0.063** | 0.594±0.023 | **0.767±0.022** | 0.748±0.009 | 0.639±0.099 |
| **LegoFuse (Ours) , 2 Epochs** | **0.734±0.032** | **0.626±0.046** | **0.863±0.112** | **0.634±0.010** | **0.771±0.020** | **0.759±0.041** | 0.701±0.023 |

Table 3: Task performance of uni- and multimodal models across 7 medical datasets – for each task target metric, we highlight the **best** and **second-best** models.

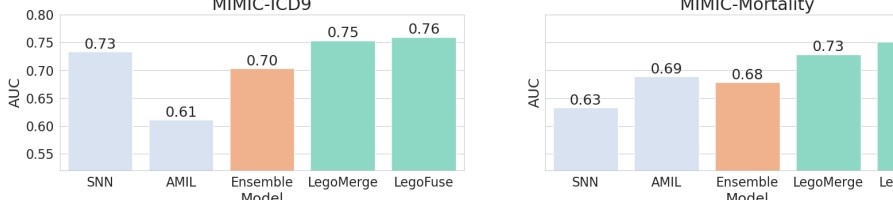

Figure 2: AUC performance on the MIMIC dataset when merging existing encoders (SNN for tabular, AMIL for Time Series) using **LegoMerge** and **LegoFuse**. Our multimodal model merge shows much better performance than using an **ensemble**, exhibiting the performance gains, at no additional costs, through the merge even prior to fine-tuning in **LegoFuse**.

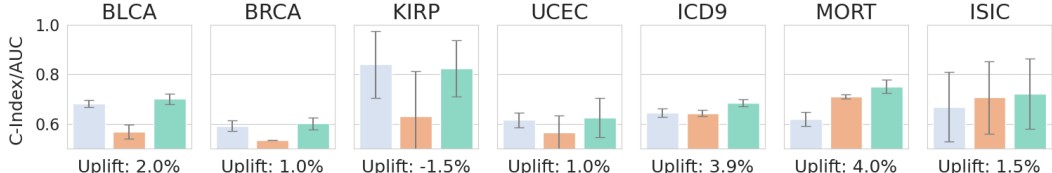

Figure 3: Mean task performance (concordance Index/AUC) of **LegoBlock (Tabular)**, **LegoBlock (Image/Time Series)** and **LegoMerge**, showing the increase in task performance by applying a multimodal model merge *without any fine-tuning*. Our proposed multimodal model merge shows a positive performance improvement on 6 out of 7 datasets.

## C   Datasets

We evaluate MM-Lego (*LegoMerge* and *LegoFuse*) and its components (*LegoBlock*) on seven multi-modal medical datasets covering three separate modalities (images, tabular, time series) from three separate sources: histopathology (The Cancer Genome Atlas (TCGA)) [39], intensive care data (Medical Information Mart for Intensive Care (MIMIC)) [40], and skin imaging (Society for Imaging Informatics in Medicine & International Skin Imaging Collaboration (SIIM-ISIC)) [41].

**TCGA**: Some of the results shown in this paper here are based upon data generated by the TCGA Research Network: `https://www.cancer.gov/tcga`. The Cancer Genome Atlas (TCGA) is an open-source genomics program run by the United State National Cancer Institute (NCI) and National Human Genome Research Institute, containing a total of 2.5 petabyts of genomic, epigenomic, transcriptomic, and proteomic data. We predict survival of right-censored patients based on the high-resolution histopathology slides ($\sim 80,000 \times 80,000$ pixels) and multi-omic data (gene expressions, copy number variations and gene mutations) captured from bulk sequencing in a tabular format. We train on four separate cancer cohorts with multimodal data available: Urorethelial Bladder Carcinoma (BLCA, $n = 436$), Breast Invasive Carcinoma (BRCA, $n = 1021$), Kidney Renal Papillary Cell Carcinoma (KIRP, $n = 284$), and Uterine Corpus Endometrical Carcinoma (UCEC, $n = 538$).

**MIMIC-III**: We train models on two separate tasks: patient mortality (multi-class classification) and disease classification (ICD-9 codes), which we formulate as a binary classification task. We use both clinical variables and small time series data on various vital signs measured at 24 time steps. Both tasks have $n = 32616$ and the same feature set for different task labels.

**SIIM-ISIC**: Stems from the Society for Imaging Informatics in Medicine & International Skin Imaging Collaboration (SIIM-ISIC) melanoma classification Kaggle challenge [41], which contains both tabular data and images of skin leisures to be classified for melanoma patients. To account for class imbalance, we randomly downsampled the majority class to a 5:1 ratio for the class of interest (melanoma) to a sample size of $n = 2875$. All images were patched and encoded using the resnet50-kather100k for TCGA (ResNet pre-trained on a large histopathology patch collection) [45] and a regular ImageNet v2 pre-trained ResNet for the pictures of skin leisures. Both images (patch encodings) and times series were represented as 2D tensors, and the tabular clinical and multi-omic data as 1D tensors to pass into the modality-specific encoders $g(\cdot)$.

## D   Losses and Metrics

The results report the (unseen) test set performance, by evaluating the concordance Index (c-Index) in the case of TCGA, AUC in the case of MIMIC-III-ICD9 and ISIC, and Macro-AUC ("one-vs-rest") for MIMIC-III-ICD9. As indicated in Figure 1 the output of each task head in $\mathcal{Y}$ are the logits with predictions for each class given the final Fourier-transformed latent state $y_l = f(L_T^{\mathcal{F}})$. Since TCGA is a survival prediction task with right-censored data, we have divided the survival period into four non-overlapping bins and use the logits of these bins to calculate the hazard ($y_h = \frac{1}{1e^{-y_l}}$) and survival ($y_s = \prod_1^k 1 - y_h$) respectively for $k$ bins. Given the hazards, censorship, and ground truth bins, we can calculate the negative log-likelihood loss from a proportional hazards model [46] which is used as the survival loss. We evaluate the performance using the Concordance Index (c-Index), for which we determine the fraction of paired samples in which the prediction outcomes are concordant with the ground truth. As MIMIC and ISIC relate to classification tasks, we employ categorical cross-entropy loss for training. Note that both AUC and the c-Index have similar interpretations, therefore the values range between $[0.5 - 1]$.

## E   Implementation Details

**Baselines.** For all experiments, we compare *LegoMerge* and *LegoFuse* to several uni- and multimodal baselines to evaluate their performance. For all tabular modalities, we use a self-normalising network [42] due to its performance and regularisation mechanisms suitable for high-dimensional tabular data. For the image and time series modalities, we use an attention-based Multiple Instance Learning (AMIL) [43]. Across all modalities, we benchmark two related iterative-learning architectures: MultiModN [32] and Perceiver [31] which generally shows strong performance across a wide range of unimodal tasks. In terms of specific multimodal baselines, we use two late fusion combinations of

SNN+AMIL, namely concatenation of their final latent representation and bi-linear fusion [47]. For the Perceiver, we use the same multimodal setup as suggested in the original paper, i.e., concatenation of modalities before passing them into the model. We use two additional domain-specific multimodal baselines: the Hybrid Early-Fusion Attention Learning Network (HEALNet) [33] which is using an end-to-end trained iterative cross-attention architecture and the Multimodal Co-Attention Transformer (MCAT) [44] which is using the tabular (1D) modality as context for the imaging (2D) modality.

**Validation & Compute.** For each experiment and dataset, we perform a 5-fold repeated random sub-sampling with a 70-15-15 train-test-validation split. We re-ran all of the baseline models in this paper using their open-source code to ensure that no performance differences are caused by different task setups, losses, or training splits. We ran a brief Bayesian Hyperparameter search [48] for key parameters of each model (learning rate, decay, schedulers, dropout, layer dimensions). The experiments were run on a single Nvidia A100 80GB GPU on a Ubuntu 22.04 virtual machine. Both the complete MM-Lego experimental code as well as its "lightweight" PyTorch package (installable via the Python Package Index (PyPI)) will be published on GitHub.

# F    Signal Interference on Latent Variables

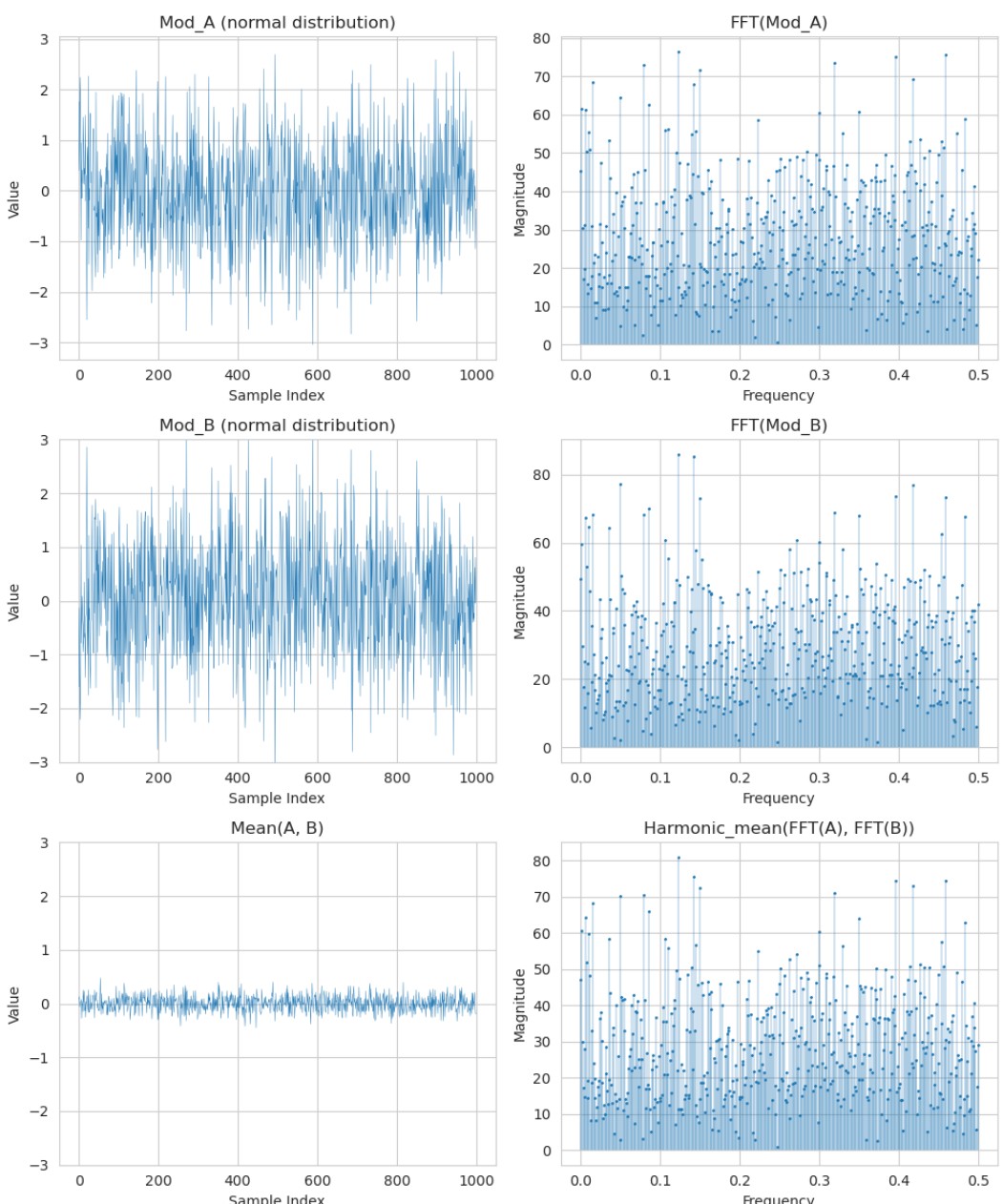

Figure 4: Example of signal interference on a random normal latent variable and its additive inverse variable with some added noise, showcasing a severe case of signal interference where nearly all signal cancels out. We can see that the fourier-transformed data does not suffer this problem when we apply the harmonic mean. This is a key reason for the choice of model merging architecture.

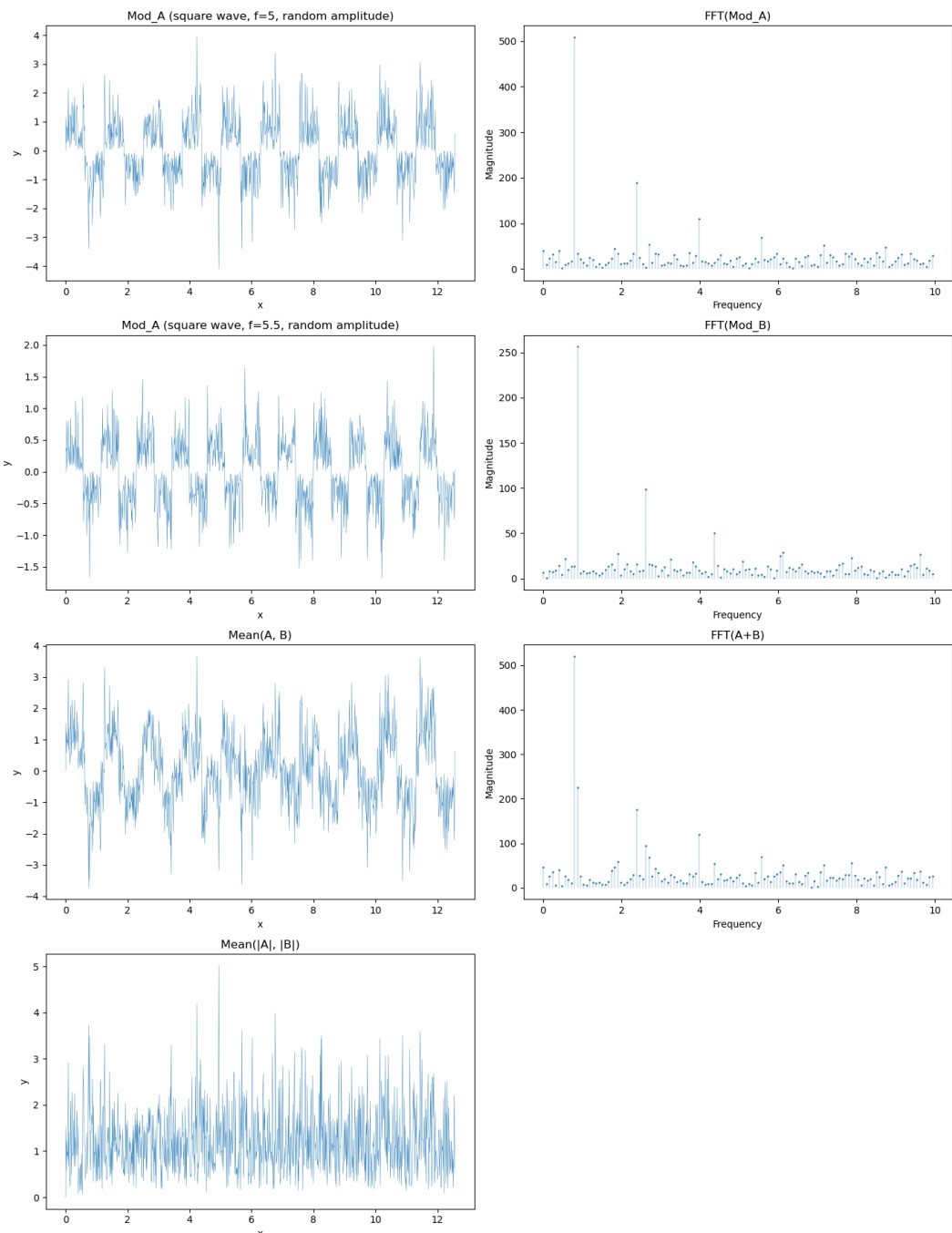

Figure 5: The argument against Fig. 4 would be to use absolute or only positive values. This example shows that this logic can also be flawed. We demonstrate this using a squarewave function with a frequency offset beteen $Mod_A$ and $Mod_B$ and a scaled amplitude by a normal distribution. We can see that the mean of the regular and the absolute values suffers some signal interference while the FFT aggregation does not.

# G   Training on unpaired data

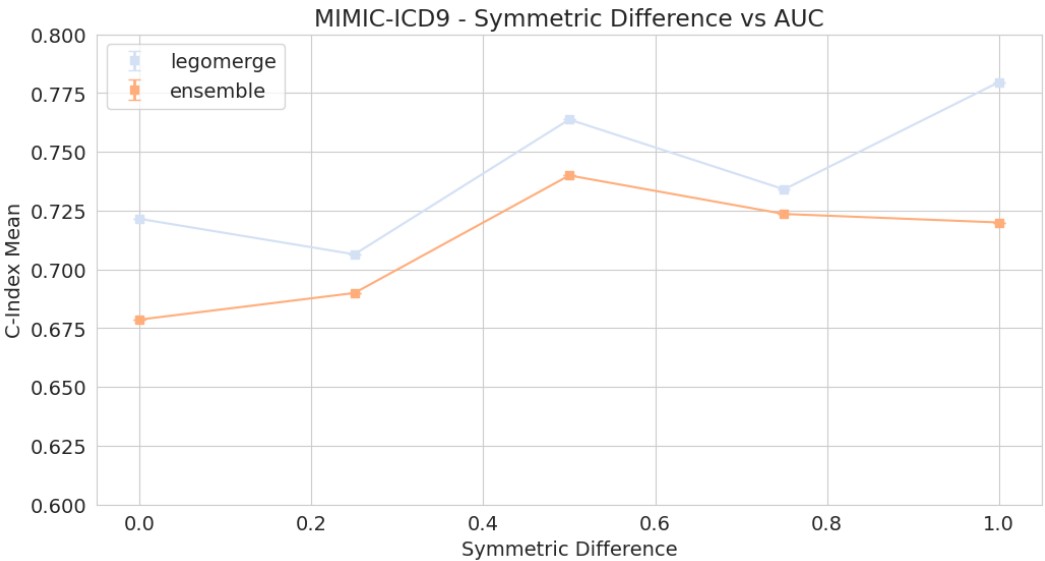

Figure 6: Test performance of *LegoMerge* (SNN+AMIL) compared to the SNN-AMIL ensemble when training on different levels of overlapping samples between the modalities. A symmetric difference of 1 means no overlap between the samples, 0 being perfect overlap. We selected N=10,000 MIMIC examples for this experiment.

