# OpenReview forum: "Multimodal Lego: Model Merging and Fine-Tuning Across Topologies and Modalities"
_NeurIPS.cc/2024/Workshop/UniReps — UniReps_

### Official Review · Reviewer_M6sd · 2024-10-05
**Multimodal Lego: Model Merging and Fine-Tuning Across Topologies and Modalities**

**Rating:** 8
**Confidence:** 2

**Review:**

The authors propose and evaluate a model merging framework, where they use the frequency domain to merge unimodal models trained separately into a well-performing multimodal model. Because they merge latent representations in the frequency domain, the framework can deal with different input shapes. They also evaluate a multimodal model that allows the modalities to interact with each other through minimal fine tuning of a stacked or layer-woven combined model and find that this model outperforms other multimodal (trained end-to-end) models on biomedical multimodal tasks.

Pros:

- Addresses a challenging problem: combining differently shaped inputs in a multimodal model
- Interesting, well-motivated, and scalable method to merge model representations in the frequency domain

Cons:

- I do not have enough experience in this area to identify any cons, if there are any

---

### Official Review · Reviewer_eTjQ · 2024-10-07

**Rating:** 8
**Confidence:** 3

**Review:**

The paper focuses on the problem of multimodal fusion and alignment, and proposes a method to fuse data from different modalities based on unimodal encoders with no or minimal fine-tuning. The method outperforms baselines in the multimodal biomedical benckmarks.

Strengths:
- The proposed method is able to make good use of unpaired unimodal data and achieves multimodal fusion with little fine-tuning.
- The idea to utilize the representations in the frequency domain is interesting.
- The method performs well in real-world multimodal datasets.

Limitations:
- Are there theoretical guarantees for the effectiveness of such fusion in the frequency domain, especially in the fine-tuning free setting?

---

### Official Review · Reviewer_uhV1 · 2024-10-07
**Review of "Multimodal Lego: Model Merging and Fine-Tuning Across Topologies and Modalities"**

**Rating:** 7
**Confidence:** 3

**Review:**

This paper presents Multimodal Lego (MM-Lego), a novel framework designed to merge unimodal encoders into a competitive multimodal model with minimal or no fine-tuning. MM-Lego addresses key challenges such as handling modality imbalance, ensuring topology-agnostic fusion, and avoiding signal interference through frequency-domain learning. The paper demonstrates impressive performance across seven medical datasets spanning multiple modalities. However, the complexity of the method’s explanation could benefit from more clarity and better visual representation of some core ideas.
The quality of the research is high, with a well-thought-out approach that addresses several significant challenges in multimodal learning, particularly in medical applications. The use of Fourier transforms to reduce signal interference is an innovative solution, and the experimental results are promising.

Strengths:

The architecture is versatile, capable of combining multiple data modalities without retraining, which is a major advantage in computationally constrained environments.
The performance improvements shown in various datasets demonstrate the method’s applicability across tasks like survival analysis, disease classification, and image-based diagnostics.
The results indicate that LegoMerge achieves comparable performance to end-to-end training models, while LegoFuse outperforms these models with just a few epochs of fine-tuning, highlighting the efficiency of the approach.
Weaknesses:

The mathematical formalism could be explained more clearly, particularly for the transformation between spatial and frequency domains. Readers unfamiliar with Fourier transforms might find these sections challenging to follow.
Although the performance is strong, additional comparisons with other state-of-the-art methods beyond the selected baselines (e.g., transformers specifically designed for multimodal data) could provide a more comprehensive evaluation of the model’s advantages.
Clarity
The paper is generally clear but could improve in certain sections, particularly in explaining some of the more technical details.

Strengths:

The figures are helpful, especially in illustrating the workflow of MM-Lego, and the architecture is well-documented in terms of its components.
The discussion of signal interference and the use of the harmonic mean in the frequency domain is insightful, providing a strong rationale for the design choices.
Weaknesses:

The explanation of LegoBlocks and how they interact with each other could be more intuitive. The reader may benefit from more visual aids or examples of how unimodal models are transformed and fused within the MM-Lego framework.
More explicit examples of real-world scenarios where the model’s flexibility and efficiency would be critical could help contextualize its practical benefits.
Originality
The originality of the approach is evident, particularly in the use of frequency-domain learning for multimodal model merging. This paper stands out by offering a flexible, scalable method that doesn’t require paired modalities for training.

Strengths:

The concept of using Fourier transforms to handle signal interference and ensure smooth model merging is novel and effective.
MM-Lego’s ability to operate on unpaired data and still achieve strong performance is a unique feature that sets it apart from many existing multimodal approaches that rely heavily on paired data.
Weaknesses:

While the approach is novel, some components, such as the fine-tuning with LegoFuse, bear similarities to standard model fusion techniques. More emphasis could be placed on how these elements differ from traditional methods.
Significance
The significance of this work lies in its potential to simplify the process of building multimodal models in fields such as healthcare, where different data modalities are common, but training paired models can be difficult or infeasible.

Strengths:

The ability to merge unimodal models without end-to-end training is a significant breakthrough, especially for large-scale medical datasets where computational resources are often limited.
MM-Lego’s flexibility and scalability make it well-suited for real-world applications, particularly in medical and scientific research, where datasets often include unpaired modalities.
Weaknesses:

The practical implications of the model’s use in real-time clinical settings or other high-stakes applications could be further explored. Demonstrating how the framework handles challenges like real-time inference or highly imbalanced datasets would add to its significance.
Pros
Innovative use of frequency-domain learning to prevent signal interference in multimodal model merging.
Flexible framework that allows for the merging of models without requiring paired modality data or retraining.
Strong performance on multiple datasets across different modalities, demonstrating the framework’s versatility.
Efficient architecture that achieves competitive results with minimal fine-tuning.
Cons
Technical explanations, particularly regarding Fourier transforms and signal interference, could be clearer for a broader audience.
Limited comparisons with more advanced multimodal models could reduce the impact of the results.
The practical benefits in real-world, time-sensitive applications need further exploration.

---

### Official Review · Reviewer_Btv4 · 2024-10-07
**Multimodal Lego: Combining Unimodal Models in Biomedical Models - Well Motivated but Lacking Clarity on the Claims**

**Rating:** 7
**Confidence:** 3

**Review:**

## Overview

This paper introduces Multimodal Lego (MM-Lego), a framework for combining unimodal models into performant multimodal models with minimal or no fine-tuning. The method is designed to address challenges in multimodal learning for biomedical tasks, where data from different modalities may have different distributions, scales, and cardinalities. The method is well-designed and has a strong motivation, but could benefit from some clarification.

## Pros

1. **Novel approach**: MM-Lego offers a unique solution to multimodal fusion that doesn't require end-to-end training.

2. **Flexibility**: The framework can work with various unimodal encoders and handle different modalities (images, tabular data, time series).

3. **Comprehensive evaluation**: The authors test their approach on seven multimodal medical datasets.

4. **Clear motivation**: The paper provides a compelling motivation for the biomedical domain application.

5. **Innovative technique**: The authors make innovative use of Fourier transforms for multimodal fusion. Page 3 describing the Fourier motivation, the cross attention, and the merging is well written.

## Cons

1. **Limited applicability to pre-existing models**:
   - The method requires training the unimodal Fourier block from scratch for each modality.
   - This requirement may limit its applicability to pre-existing models.
   - The abstract and introduction initially suggest that any pretrained unimodal model can be used without additional training, which does not seem to be the case. Only the specific unimodal lego black can be used without additional training.

2. **Imprecise notation**:
   - In line 47, is the union of $\cup_{m \in \mathcal{M}} m$ just $\mathcal{M}$? Did the authors mean $x^{m}$?
   - The notation $\mathcal{M}$ from A to Z implies precisely 26 modalities.
   - In line 50, is it only $A \in \mathbb{R}^{I \times J \times K}$, or is any modality $m \in \mathbb{R}^{i \times j \times k}$?
   - What is $S$ in equation (1)? Is $S = \mathbb{R}$ or more precisely $S = \mathbb{Z}^+$ (set of positive integers)?
   - In line 60, what is $h_m$? This variable appears to be undefined.

4. **Vague comparisons in Table 1**:
   - The table is neat but somewhat vague/subjective.
   - Are the authors referring to specific previous works?
   - For Late Fusion, can't three separate unimodal models perform without end-to-end training or add modalities without retraining by averaging the output logits or performing simple voting for the output class?
   - The meaning of "architecture agnostic" in the context of this table needs clarification.
   - Don't the Lego methods require that the architecture extracts a latent vector, thus not any architecture can be used for them?
   - Or does this refer to the fact that only the feature extractor $g$ can be implemented with various backbones?

5. **Figure 1 clarification**:
   - Should the task head in (1) be outside of the Fourier Lego block?
   - If not, in 2A, would the authors be taking the harmonic mean of the output logits?
   - In 2B, would they be passing the output logits into the next block?

6. **Highlight disjoint data experiments**:
   - In line 62, the authors provide a good motivation for single-cell data versus bulk sequencing data and the need to train on different samples for the same task.
   - Did any of the tested datasets have disjoint data like this for different modalities?
   - If so, this should be highlighted as it appears to be the main motivation for medical data applications. It's not clear whether this goal is achieved by the end of the paper.

---

> ### Author Response · Authors · 2024-11-04
>
> Thank you very much for your thoughtful, detailed, and constructive feedback. We are very encouraged by your positive view of the paper. We acknowledge the notational clarifications and have addressed these along with other comments in the camera-ready version.
>
> We also would like to answer some of your questions below and encourage you to reach out on the day of the workshop to discuss this further!
>
> **Comments on Table 1**:
> * Vagueness/subjectivity: We had to make some compromises given the space constraints of the extended abstract format which meant that we briefly cut an introductory section to the late, intermediate, and early fusion benchmarks. We have used a part of the additional page in the camera-ready version for these clarifications.
> * Late fusion: It's true that you could use simple ensembling, but the performance is typically worse than the unimodal model, as we show in Appendix B. Hence the wording of "_performant_ without end-to-end-training"
> * Architecture agnostic: this refers to 1) that many model merging methods require equivalent architectures (e.g., in multi-task settings) while MM-Lego does not make that assumption and 2) that you can use it with any modality-specific encoder
>
> **Comments on Figure 1**:
> * Task head: Yes, the task head is outside of the Lego block and we have updated the grey box in Fig. 1.1 to reflect that. This should also answer the remaining questions.
>
> **Experiments and relevance**
> * The experiments are still run on paired datasets to allow a fair comparison with the fusion baselines, which require paired data. We agree that this would be an interesting additional experiment which can be conducted with TCGA where we have a lot of disjoint data available for other modalities that we currently do not include. It is worth noting that the fact that we "artificially" disjoin the data for the LegoMerge and LegoFuse experiments, so the experiments shown here are still relevant to its core motivation.

---

### Official Review · Reviewer_GXpz · 2024-10-07

**Rating:** 8
**Confidence:** 3

**Review:**

**Paper Summary** \
The paper introduces a novel framework (Multimodal Lego) for merging unimodal encoders into a multimodal model using model-specific wrappers. Standard modality merging frameworks require paired data for jointly training the encoders end-to-end, which poses a significant challenge in practice. This work, however, enables training encoder-specific wrappers in isolation, requiring no paired data or homogeneous encoder architectures, which is often assumed in the model merging literature. These wrappers, referred to as Lego Blocks, exploit the advantages of the frequency domain, such as robustness to aggregation interference, to generate modality-agnostic and shape-consistent representations, facilitating merging. The paper proposes two variants of merging: one without end-to-end fine-tuning and one with, where the latter achieves state-of-the-art performance in their benchmarks. The use of frequency-domain representations, often overlooked, and the ability to merge encoders with arbitrary architectures are key contributions of this work.

**Strengths** \
Lego Merging is highly flexible and practical, as it requires no paired data and allows the use of arbitrary encoder architectures. The decision to use the frequency domain appears well-grounded, and the merging process is efficient. Additionally, the design of Lego Blocks significantly reduces the engineering effort for aligning different modalities, as they produce compatible representations. Overall, these strengths make the framework a notable contribution to the field.

**Weaknesses** \
The motivation for the iterative updates within each block is not fully explained. It would be helpful to clarify how these iterations contribute to the learning process and whether they affect convergence or model stability. Furthermore, the paper does not adequately discuss the computational costs associated with multiple iterations, which may be a concern.

**Questions & Suggestions**
 - Clarify the motivation behind the iterative updates and how they affect model performance and stability. Also, provide details on the computational overhead associated with these iterations.
 - Consider including additional datasets from non-medical domains (e.g., vision-and-language or audio-visual tasks) to demonstrate the generalizability of the framework across diverse modalities. Replacing some results from the medical domain with results from new domains could broaden the paper's applicability.
 - Given the flexibility of the framework, it would be beneficial to include ablation studies on key design choices, such as the use of frequency domain vs. spatial domain representations, and the effect of stacking vs. weaving layers in the LegoFuse variant. This would significantly strengthen the claims of generalizability and adaptability of the Multimodal Lego framework.

---

> ### Author Response · Authors · 2024-11-04
>
> Thank you very much for your thoughtful and constructive feedback. We are very encouraged by your positive view of the paper and address your questions below as well as the camera-ready version:
>
> **Motivation for iterative updates**: The main motivation behind an iterative architecture is that missing modalities (or "blocks") can be easily skipped for individual samples, which is in line with the original motivation of the paper that little paired data is available. We will make sure that this is clearer in the methods section of the final manuscript.
>
> **Adding datasets and additional ablations**: We fully agree with this suggestion as no current methodological choices would restrict the method to medical modalities. This is planned for a further iteration of this project.

---

### Author Response · Authors · 2024-11-04
**Thank you for your reviews**

We would like to thank all reviewers for their constructive feedback on our extended abstract submission! Please do not hesitate to come and talk to us during the workshop to discuss some of these comments further!

---

### Decision · Program_Chairs · 2024-10-10

**Decision:**

Accept (Oral)

**Comment:**

In light of the positive reviewers' feedback and relevancy of the submission, we are pleased to accept this paper for presentation at UniReps 2024. We kindly ask the authors to incorporate the reviewers' suggestions and feedback in the final camera-ready version of the manuscript.